# Effect of Atomized Black Maca (*Lepidium meyenii)* Supplementation in the Cryopreservation of Alpaca (*Vicugna pacos)* Epididymal Spermatozoa

**DOI:** 10.3390/ani13132054

**Published:** 2023-06-21

**Authors:** Gloria Levano, Juana Quispe, Diego Vargas, Marlon García, Alberto López, Luis Aguila, Martha Valdivia

**Affiliations:** 1Laboratory of Reproductive Physiology, Biological Sciences Faculty, Universidad Nacional Mayor de San Marcos, Lima 15081, Peru; gloria.levano1@unmsm.edu.pe (G.L.); juanalady.quispe@unmsm.edu.pe (J.Q.); diego.vargas1@unmsm.edu.pe (D.V.); 2Laboratory of Genetics, Biological Sciences Faculty, Universidad Nacional Mayor de San Marcos, Lima 15081, Peru; marlon.garcia@unmsm.edu.pe (M.G.); alopezs@unmsm.edu.pe (A.L.); 3Laboratory of Reproduction, Centre of Reproductive Biotechnology (CEBIOR-BIOREN), Universidad de La Frontera, Temuco 4811322, Chile

**Keywords:** freezing medium, sperm function, antioxidants, alpaca sperm, small ruminants

## Abstract

**Simple Summary:**

Successful cryopreservation of semen of South American camelids has been limited, hindering the application of artificial insemination. In this scenario, the addition of antioxidants to semen extenders provides a strategy to improve sperm freezability. Thus, the objective of this study was to evaluate the effect of the addition of black maca in the freezing medium of epididymal spermatozoa of alpaca species. Fifteen pairs of epididymis were collected from a local slaughterhouse. Each sample was divided into six groups: (1) fresh, (2) yolk medium (YM), (3) 10 mg/mL maca, (4) 20 mg/mL maca, (5) 30 mg/mL maca, and (6) resveratrol (as an antioxidant control). Sperm cryopreservation was performed through the slow freezing method. The sperm motility, viability, plasma membrane integrity, DNA integrity, total ROS production, and mitochondrial function were analyzed. The results show that maca (20 mg/mL) improved the parameters associated with sperm function. Similarly, ROS production decreased, although DNA integrity was similar among the groups. These results suggest that maca at 20 mg/mL has cytoprotective effects during freezing/thawing of epididymal sperm of alpaca species. Further research will be focused on assessing the effects of maca supplementation on semen extenders by using biomolecular markers associated with fertility.

**Abstract:**

Artificial insemination is an important assisted reproductive technology that has been applied in several mammalian species. However, successful cryopreservation of semen of South American camelids has been limited, hindering the commercial application of artificial insemination in alpaca species. In this scenario, the addition of antioxidants to semen extenders provides a strategy to improve the freezability of mammalian sperm. Bioactive metabolites from natural extracts of black maca have shown valuable antioxidant properties. Thus, the objective of this study was to evaluate the effect of the addition of atomized black maca in the freezing medium of epididymal spermatozoa of alpacas. Fifteen pairs of epididymis were collected from a local slaughterhouse. Each sample was divided into six groups: (1) fresh, (2) yolk medium (YM), (3) 10 mg/mL maca, (4) 20 mg/mL maca, (5) 30 mg/mL maca, and (6) resveratrol (as an antioxidant control). Sperm cryopreservation was performed through the slow freezing method. Markers associated with functionality, such as motility, viability, and plasma membrane integrity, as well as markers associated with oxidative damage, such as DNA integrity, total ROS production, and mitochondrial function, were analyzed. The results show that the supplementation with black maca (20 mg/mL) improved the sperm motility, viability, plasma membrane integrity, and mitochondrial function evaluated according to an index of formazan deposits. Similarly, the ROS production decreased with maca at 20 mg/mL, although the DNA integrity did not show any differences among the groups. These results suggest that maca at 20 mg/mL has cytoprotective effects during freezing/thawing of epididymal sperm of alpaca species. Further research will be focused on assessing the effects of maca supplementation on semen extenders by using biomolecular markers (proAKAP4) associated with fertility.

## 1. Introduction

Peru is the world’s leading producer of alpacas (*Vicugna pacos*) and the second-leading producer of llamas (*Lama glama*) [1]. Due to the economic importance that these species represent, several groups have focused the on the conservation and freezing of semen of South American camelids in Peru. Specifically, the alpaca species has been used as a model for cryopreservation of semen, in vitro fertilization (IVF), and artificial insemination; however, the expected outcomes have not been reached, such as in other domestic species [2].

Gamete freezing, specifically cryopreservation of mammalian spermatozoa, is an important tool in assisted reproduction, safeguarding the species’ genetic pedigree; it allows long-term preservation of genetic resources and/or preservation of endangered species [3]. Nevertheless, cryopreservation of alpaca spermatozoa has not been used extensively because of the limiting step of semen collection and poor freezability being associated with specific semen characteristics [4], such as a low sperm count, a high percentage of abnormalities, and the viscosity of seminal plasma [5].

During cryopreservation, the sperm are exposed to different stressors, such as the formation of ice crystals, toxicity of the cryoprotectants, and changes of temperature during chilling, freezing, and rewarming [6]. These changes lead to osmotic and thermal shock and oxidative stress, which are among the main biophysical factors affecting sperm viability [7]. Oxidative stress refers to the increase of intracellular concentrations of reactive oxygen species (ROS) [8]. Spermatozoa are susceptible to oxidative stress through all semen processing, which is why it is considered one of the main factors associated with fertilization failures [9].

Nevertheless, controlled ROS production is essential for activation of sperm motility, capacitation, and fertilization [10]. During a pathophysiological imbalance, however, ROS accumulate and react with biomolecules (i.e., membrane phospholipids, enzymes, and chromatin), impairing the structure and cellular functions [8,11]. ROS can induce lipid peroxidation and DNA fragmentation and affect sperm motility and their ability to fertilize [10]. Lipid peroxidation associated with OS decreases sperm motility and viability, acrosome integrity, and mitochondrial function [12].

However, it has been shown that the detrimental effects of ROS imbalance can be partially prevented by adding antioxidants during in vitro manipulation of spermatozoa [13]. Therefore, antioxidant therapy with the addition of antioxidants to semen extenders has been an effective strategy for improving the quality of frozen-thawed sperm [14]. For instance, methionine, vitamin C (ascorbic acid), and vitamin E (α-tocopherol) have been used in semen extenders to protect the cells against cryoinjury [15]. Nevertheless, dilution or cooling of these antioxidants during the cryopreservation process could reduce their benefits due to an antioxidant defense.

In this scenario, a natural source of additives to preserve and enhance sperm function during semen storage is plant extracts [16]. Plants produce a wide range of chemical compounds with bioactive properties, including alkaloids, phenolic compounds, and terpenoids. For instance, resveratrol, a natural polyphenolic compound, has been extensively used as a suitable antioxidant supplement for semen extenders in human and domestic species [17]. Furthermore, these natural metabolites possess antimicrobial activities and act as ROS scavengers. Therefore, plant extracts represent a good alternative to conventional antioxidants used for the cryogenic storage of semen.

Maca (*Lepidium meyenii* Walpers or *Lepidium peruvianum* Chacón), from the Brassicaceae family, is a native Peruvian plant that grows in the central Andes between 4000 and 4500 m above sea level. The root of this plant has been used as folk medicine to increase fertility due to its high sucrose content [18] and phytochemicals [19,20,21]. Several health benefits have been reported about its bioactive compounds, such as improving sexual performance in animals [22] and in humans [23], spermatogenesis [24,25], energizing anti-stress and antioxidant effects [26,27,28], natural cytoprotection [29,30], and cytoprotection against oxidative stress conditions [26,31]. These effects have been attributed to a high concentration of antioxidants and other secondary metabolites that help maintain low ROS levels by increasing the enzymatic activity of superoxide dismutase and through the direct elimination of free radicals [29,30]. Along this line, recent studies have suggested the use of maca as an alternative supplement for the preservation of semen quality in different reproductive biotechnologies, such as for sperm storage and in vitro fertilization (IVF) [31,32,33].

Therefore, regarding the valuable cytoprotective and antioxidant capacities of maca, the purpose of this study was to evaluate, for the first time, the effects of atomized extract of maca during cryopreservation of epidydimal alpaca (*Vicugna pacos*) spermatozoa.

## 2. Materials and Methods

### 2.1. Sample Collection

Fifteen pairs of alpaca (*Vicugna pacos*) testicles and epididymis from males between 4 and 6 years old were obtained post-mortem from the local slaughterhouse, Huancavelica (3000–3700 m.a.s.l), Peru. The samples were stored in 0.9% NaCl at 10 °C and transported for 22 h to the laboratory.

### 2.2. Obtaining the Epididymal Sperm

At the laboratory, we proceeded to isolate the spermatozoa from the cauda epididymis. The cauda epididymis was removed and cut into small pieces in 1 mL of 0.9% NaCl 9% at 37 °C. Then, once the samples were extracted, the suspension containing the spermatozoa was transferred to 1.5 mL plastic tubes and placed in an incubator to warm up to 37 °C. Subsequently, the samples were divided into two equal parts. One part was washed with PBS using centrifugation at 400× *g* for 5 min and then used to determine the motility, viability, and integrity of the plasma membrane in the raw sample. The second part was used for cryopreservation. To carry out the cryopreservation of the spermatozoa, the exclusion criterion was a motility value greater than or equal to 70% in the fresh sample.

### 2.3. Sperm Cryopreservation and Maca Supplementation

The medium used for cryopreservation was Tris yolk medium (YM) with 10% fetal bovine serum (FBS) containing 0.25 M fructose and 10% dimethyl sulfoxide. To assess the effects of maca during the cryopreservation of epidydimal alpaca sperm, the freezing medium (YM) was supplemented with the atomized hydroalcoholic (methanolic) extract of a commercial black maca powder (JUVENS^®^ Cayenatur, Lima, Peru), as described by the Research Circle of Plants with Effect on Health (Grant no. 010-2014-FONDECYT). Botanical samples were deposited in the HEPLAME MG-2015 (Herbarium of Medicinal Plants, Section of Pharmaceutical Sciences, Faculty of Sciences and Philosophy, Universidad Peruana Cayetano Heredia). Maca components have been previously characterized [34]. The atomized black maca was added in three concentrations of 10 mg/mL, 20 mg/mL, and 30 mg/mL. In addition, the supplementation of the YM freezing medium with 5 mg/mL of resveratrol (Sigma, St. Louis, MO, USA) was included as an antioxidant control group. The straws were sealed and placed in the thermo-controlled freezing system using the CRYOBATH system directed by the Cryogenesis Version 4.0 (Cryologics) software; the freezing protocol was as follows: first, cooling at 4 °C for five minutes. Second, cooling from 4 °C to 2 °C at a rate of 3 °C/min and keeping the samples at 2 °C for one minute. Third, cooling from 2 °C to −30 °C with a freezing rate of 5 °C/min, and finally from −30 °C to −80 °C with a cooling rate of 8 °C/min. After this point, the straws were placed in liquid nitrogen until thawed [35].

### 2.4. Evaluation of Sperm Parameters

#### 2.4.1. Sperm Motility

Sperm motility was evaluated subjectively according to the published guidelines of the World Health Organization [36], as previously reported by our group [34]. To do this, 10 µL of the sample was placed on a pre-heated slide at 37 °C and observed under bright-field microscopy (EUROMEX, Arnhem, The Netherlands) at a magnification of 400×. Sperm motility with forward movement was considered progressive motility. The values were expressed in percentages based on the count of 100 spermatozoa in the observed field for each sample.

#### 2.4.2. Sperm Viability

The eosin-Y staining (0.5% wt/vol) was performed by mixing 5 μL of semen with 5 μL of the stain on a pre-heated (37 °C) microscope slide, and the viability was evaluated within 2 min after addition of the stain using bright-field microscopy (EUROMEX, Arnhem, the Netherlands) at 400× magnification. The values were expressed in percentages based on the count of 100 spermatozoa in the observed fields for each sample.

#### 2.4.3. Plasma Membrane Integrity

The plasma membrane integrity was evaluated through the hypoosmotic test, also known as the HOST test. Briefly, a hypoosmotic solution was prepared with a final osmolality of 100 mOsm. Next, 30 μL of the sperm suspension was incubated with 270 μL of the hypoosmotic solution for 30 min at 37 °C. Then, the mixed solution was placed on a coverslip and observed under bright-field microscopy (EUROMEX, Arnhem, Holland) at 400× magnification. The coiling of the sperm tail representing an intact flagellar membrane was evaluated according to the guidelines reported previously [37].

#### 2.4.4. DNA Fragmentation Analysis

The evaluation was carried out using the commercial terminal deoxynucleotidyl transferase-mediated dUTP nick-end labelling (TUNEL) kit (in situ cell death detection Kit, Fluorescein Roche^®^ Mannheim, BW, Germany). The sample’s processing was performed as previously described with some modifications [38]. A sperm suspension of 2 × 10^6^ spermatozoa /mL was fixed in 4% formaldehyde for 15 min at 4 °C. Subsequently, two washes were carried out with a phosphate saline solution (PBS). Next, the samples were permeabilizated by being resuspended in 100 μL of 0.5% Triton X-100 + 0.1% sodium citrate for 30 min at room temperature. The permeabilization solution was washed twice by centrifugation in PBS at 500× *g* for 5 min.

Next, the permeabilized spermatozoa were incubated with the TUNEL reaction mixture in a dark environment at 38.5 °C for 1 h. After this time, they were washed with 500 μL of PBS and centrifugated at 500× *g* for 6 min. The pellets were resuspended in 400 µL of PBS containing 2 μL (2.4 mM stock solution) of propidium iodide (Live/Dead^®^ Sperm Viability Kit, Molecular Probes L-7011, EEUU). The samples were conserved in a dark chamber at 4 °C before being evaluated by fluorescence microscopy (EUROMEX, Zeiss Axio Scope A, Arnhem, The Netherlands). Negative and positive controls, respectively, were performed by omitting the terminal transferase enzyme and preincubating fixed and permeabilized semen samples with DNase I (1 mg/mL) for 20 min at room temperature. For TUNEL-positive spermatozoa, the green fluorescence was detected by using a 530/30 nm bandwidth filter, and the red fluorescence (for the propidium iodide stain) was detected by using a 585/42 nm bandwidth filter. For each sample, the percentage of spermatozoa depicting green fluorescence (TUNEL+) was calculated based on the count of 100 spermatozoa for each sample.

#### 2.4.5. Evaluation of ROS Production

The evaluation of the intracellular ROS was performed using the reactive oxygen species assay kit (Beyotime Institute of Biotechnology), following the manufacturer’s instructions. The 2’,7’-Dichlorofluorescein diacetate (H₂DCFDA) is a cell-permeable, non-fluorescent reagent deacetylated by cellular esterase to a non-fluorescent 2’,7’-dichlorofluorescein (H₂DCF) after diffusion into the cell. ROS then oxidizes the H₂DCF to a highly fluorescent 2’,7′-dichlorofluorescein (DCF). After staining, the samples were kept in a dark chamber and evaluated by fluorescence microscopy (EUROMEX, Zeiss Axio Scope A, The Netherlands). Fluorescence was detected by using a 530/30 nm bandwidth filter [39]. The results are expressed as the percentage of cells positive to H₂DCFDA (ROS+) staining based on the count of 100 spermatozoa for each sample.

#### 2.4.6. Sperm Mitochondrial Activity

The mitochondrial activity was analyzed using the nitroblue tetrazolium (NBT) test for a cytochemical reaction of oxidoreductases according to Torres, et al. (2019). Briefly, a 10 μL drop of sperm suspension was mixed with 10 μL of 1 mg/mL of NBT on a pre-heated slide to 37 °C. The oxidation–reduction system was placed in an anaerobic chamber for 45 min at 37 °C. The spermatozoa were evaluated under bright-field microscopy (EUROMEX, Arnhem, Holland) with the 100× oil immersion objective. The mitochondrial activity and the functionality index for each sample were assessed based on the classification of the formazan deposit, which was determined according to the classification in percentages, and obtained from the number of spermatozoa classified in each category (compact: “C”, sub-compact: “SC”, focal: “F”, residual: “R”) and their corresponding factor (1, 0.7, 0.3, 0.1, respectively) proposed by Hrudka [40]. The cytochemical activity rates were determined by using a semi-quantitative rating according to the amount of formazan deposited in the mitochondrial sheath, where the compact (strong and uniform staining) and subcompact (medium to strong deposit) both were classified as intensive reactions. On the other hand, focal (few localized deposits) and diffuse (weak and/or absent reaction) patterns were related to low and residual rates as reduced reactions [40,41].

### 2.5. Statistical Analysis

The data were analyzed using the Shapiro–Wilk test and ANOVA for distribution. The data (mean ± SD) were compared using multiple comparison *t*-tests (non-parametric test). The Wilcoxon test was used to analyze the NBT data that did not present a normal distribution. The statistical analysis and graphical representation were performed using the R version of RStudio 1.4.17173.

## 3. Results

The sperm motility, viability, plasma membrane integrity, DNA fragmentation, total levels of ROS, and index of mitochondrial activity are detailed in the Appendix A.

### 3.1. Effect of Maca (Lepidium meyenii) on Sperm Motility

First, we evaluated how the sperm motility was affected by the cryopreservation process and supplementation of the freezing medium with maca or resveratrol. The addition of 10 mg/mL and 20 mg/mL of maca in the freezing medium improved the sperm motility compared to the control group (YM) (Figure 1; Appendix A). The sperm motility with maca at 10 mg/mL and 20 mg/mL was also greater (*p* < 0.05) than the group supplemented with resveratrol (Figure 1; Appendix A).

### 3.2. Effect of Maca (Lepidium meyenii) on Sperm Viability

The cryopreservation process also affected sperm viability (Figure 1; Appendix A). In addition, 20 mg/mL of maca or resveratrol increased viability by 48.8 ± 5.69% and 50.0 ± 5.64%, respectively, when compared with the control YM group (43.8 ± 4.83%) (*p* < 0.05). After thawing, the spermatozoa cryopreserved in YM supplemented with 20 mg/mL showed similar viability to the group supplemented with resveratrol (*p* > 0.05). Finally, fresh semen had the highest proportion of viable sperm (61.25 ± 8.29%) (Figure 1; Appendix A).

### 3.3. Effect of Maca (Lepidium meyenii) on the Plasma Membrane Integrity

Next, the HOST test showed that cryopreservation significantly affected the membrane integrity at 69.16 ± 8.21% vs. 51.25 ± 7.72% for fresh and cryopreserved spermatozoa, respectively (*p* < 0.05). Moreover, 20 mg/mL of maca increased the viability after thawing (58.33 ± 7.78%) compared to the control Tris yolk medium (51.25 ± 7.72%) (*p* < 0.05), showing similar levels to the group supplemented with resveratrol (57.5 ± 3.37%) (Figure 1; Appendix A).

### 3.4. Effect of Maca (Lepidium meyenii) on Reactive Oxygen Species (ROS)

The proportion of ROS+ spermatozoa increased after cryostorage in YM (40%) with respect to the initial pre-freezing (fresh) samples (29.2 ± 10.7%) (*p* < 0.05) (Figure 2 and Appendix A). The groups treated with 10 and 30 mg/mL of maca and resveratrol showed a similar level of ROS+ cells (range 22–30%) as the fresh samples (29.2 ± 10.7%) (Figure 2 and Appendix A). In addition, there was a significant decrease in the proportion of ROS+ sperm among all the groups of maca and resveratrol compared to the yolk-medium group (40 ± 12.6%) (Figure 2 and Appendix A). Furthermore, the supplementation with 20 mg/mL of maca showed the lowest (*p* < 0.05) proportion of ROS+ cells (19 ± 7.33%) compared to all the experimental groups (Figure 2 and Appendix A).

### 3.5. Effect of Maca (Lepidium meyenii) on Sperm DNA Fragmentation

DNA fragmentation increased after cryopreservation. For instance, the percentage of TUNEL+ sperm increased from 1.6% before freezing (fresh) to around 4% after freezing in the yolk medium (Appendix A). No significant differences (*p* > 0.05) were observed among the groups after freezing/thawing (Figure 3 and Figure 4; Appendix A).

### 3.6. Effect of Maca (Lepidium meyenii) on Mitochondrial Activity Index

The mitochondrial activity and functionality indexes for each sample were assessed based on the classification of formazan deposits (Figure 5) obtained from the number of cells classified in each category (compact: “C”, subcompact: “SC”, focal: “F”, and residual: “R”). No significant differences (*p* > 0.05) were found among all the groups (Appendix A, Figure 5 and Figure 6).

### 3.7. Effect of Maca (Lepidium meyenii) on Cytochemical Activity of Oxidoreductases

The cytochemical activity rates were identified according to the amount of formazan deposited in the mitochondrial sheath (Figure 6), where the compact (strong and uniform staining) and subcompact (medium to strong deposit) patterns were associated with the standard and substandard rates, respectively. Both compact and subcompact patterns were classified as intensive reactions. On the other hand, focal (few localized deposits) and diffuse (weak and/or absent reaction) patterns were associated with low and residual rates and classified as reduced reactions. The results of the comparison of cytochemical activities are summarized in Table 1. Among the treatments with maca, an intensive reaction was found with maca at 10 mg/mL (61.6%), and similar to the yolk medium (70.2%) group. Maca 30 mg/mL (51.3%) showed similar (*p* < 0.05) intensive reaction compared to the fresh sample (51%). Finally, maca at 20 mg/mL and resveratrol presented the lowest levels of intensive reaction (38% and 35%, respectively) (*p* < 0.05) (Table 1), evidencing lower mitochondrial activity.

On the other hand, the yolk medium group and the treatment with maca at 30 mg/mL showed a lower (*p* < 0.05) index of a reduced reaction (0.65% and 0.61%, respectively) compared to the fresh group (0.9%), but higher than the treatments with maca at 10 and 20 mg/mL (0.55% and 0.41%, respectively) (Table 1). The highest value (2.22%) was recorded in the resveratrol (positive antioxidant) control group. Maca at 10 mg/mL and 20 mg/mL presented the lowest indexes (0.55% and 0.41%, respectively) among the cryopreserved groups, showing a higher mitochondrial function (*p* < 0.05) (Table 1).

## 4. Discussion

The present study shows the effects of the supplementation of the freezing medium with maca (*L*. *meyenii*) on the functional parameters of cryopreserved epididymal sperm from alpaca species. In addition, an antioxidant control group supplemented with resveratrol was included as a positive control due its antioxidant properties. In this study, we evaluated several cellular features associated with sperm functionality, such as viability, plasma membrane integrity, DNA integrity, and mitochondrial function. Our data suggest that the addition of atomized black maca protects against the cryodamage at the plasma membrane and mitochondrial levels.

We are reporting, for the first time, the addition of maca in the freezing medium for the cryopreservation of alpaca (*Vicugna pacos*) sperm derived from cauda epididymis. The use of sperm from the epididymis represents useful material for the development of a protocol for the cryostorage of alpaca semen to test semen diluents and additives, such as antioxidants. Moreover, in the event of sudden death or emergency castration of a valuable sire, the use of epididymal sperm allows us to salvage valuable genetics that would have been lost otherwise [42]. Thus, cauda epididymis from castration or slaughtering can be used as a model [43]. Using this model, we found that alpaca epididymis spermatozoa undergo cryodamage after freezing/thawing, consistent with the described effects of cryopreservation in mammalian sperm physiology [44]. Cryopreservation severely affected sperm motility and viability. Similarly, the mitochondrial function was also damaged by freezing/thawing. Decreases in sperm motility may also be associated with impaired mitochondrial function after cryopreservation [12,45]. Therefore, the supplementation of the freezing medium with maca can be considered a valid strategy to reduce cryogenic damage in this species.

Studies using epididymal sperm have shown variable values of viability post-thawing (range 21 to 32%) [4,35,46], where sperm motility was the most affected parameter [35,47]. In the present study, the HOST test, which evaluates whether an intact membrane is biochemically active, was used to analyze the integrity of the plasma membrane, since the assessment of the sperm membrane function appears to be a significant marker for the fertilizing capacity of spermatozoa. Additionally, a vital staining (eosin Y) was also applied to confirm the results of the HOST test. Our data show that the addition of maca (20 mg/mL) to the freezing medium improves motility, viability, and plasma membrane integrity after thawing. In agreement, other studies have demonstrated that the supplementation of the freezing medium with black maca enhanced survival associated with optimized mitochondrial activity of spermatogonial stem cells of alpaca species [48]. Although we did not find differences in the rate of DNA fragmentation between the groups, there was a trend towards a higher rate in the cryopreserved samples. Comparable to previous reports [4,46], our results show that cryopreservation is associated with an increase in ROS production. Cryopreservation can damage sperm cells via peroxidation of the membrane lipids and production of toxic aldehydes, leading to mitochondrial damage, reduction in membrane fluidity, loss of the integrity of the plasma membranes, and damage of the sperm DNA [49], which is consistent with the observations recorded in our study. Importantly, the supplementation with maca and resveratrol also reduced this influx of ROS during cryopreservation.

This cytoprotective effect of maca can be associated with its antioxidant capacity [50,51]. The components of black maca have been extensively studied in recent years due to their antioxidant property. The presence of glucosinolates and macamides allows maca (*L*. *meyenii*) to protect cells from oxidative stress damage [52,53]. Furthermore, the presence of several secondary metabolites, such as macaridine, macamides, maca alkaloids, and glucosinolates, are typical in this plan [54]. These secondary metabolites are responsible for the biological and medicinal properties of the maca [28,50].

Different studies have shown that oral administration of maca improves seminal quality in humans [55], mice [25,56,57], and stallions [58]. For instance, the study of Rubio et al. (2006) [27] indicated that oral administration of maca to rats exposed to lead acetate (LA) prevented LA-induced spermatogenic disruption, suggesting this as a potential treatment for lead exposure-associated male infertility. Similar results were observed in horses where the dietary supplementation with maca increased sperm production and stabilized semen quality during chilled storage [59]. However, only two groups have studied the effects of maca on sperm parameters in vitro [31,32]. The study of Aoki et al. (2018) [32] found that maca improved IVF rates by optimizing the acrosome reaction and increasing sperm motility in both human and mouse models. Similarly, maca supplementation showed a general improvement in all the sperm parameters analyzed (motility, viability, acrosome integrity, morphology, and DNA fragmentation) of frozen and thawed bovine spermatozoa [31]. These data agree with our findings, wherein maca supplementation improved the sperm motility and plasma membrane integrity and viability, and decreased ROS production after thawing, showing comparable levels to the known antioxidant molecule resveratrol. In addition, it was also observed that the group supplemented with maca (20 mg/mL) and resveratrol showed less mitochondrial activity. It can be speculated that this effect may be associated with an optimized clearance of ROS, thus maintaining controlled mitochondrial activity. However, this assumption requires further investigation. It is important to note that the higher dose of maca evaluated in this study did not show beneficial effects on the parameters evaluated. In similar fashion to the results reported by Leiva-Revilla et al. (2022) [31], the adverse effect of high concentrations of maca may be associated with “hormesis”, which is an adaptive response to a stressor, such that low-dose stimulation results in beneficial outcomes, while high-dose exposure produces a toxic effect [60]. However, because controlled concentrations of ROS are necessary for proper sperm function, including fertilization capacity [58], there is also the possibility that excessive ROS clearance may be detrimental to sperm function. Therefore, it is highly recommended to determine the optimal range of concentrations of each maca extract to be used in in vitro studies.

Finally, further research will be focused on assessing the effects of maca supplementation on semen extenders by using biomolecular markers associated with fertility. For example, the sperm protein proAKAP4 is a molecular marker related to sperm motility, fertility, and freezability [60,61]. In addition, it has been shown that the proAKAP4 biomarker can be used successfully to evaluate the efficacy of the use of various extenders for the cryopreservation of bovine and ovine semen [62,63].

## 5. Conclusions

In conclusion, the supplementation with 20 mg/mL of black maca in the medium improves sperm parameters, such as sperm motility, sperm viability, and membrane integrity; it also decreases the emission of total sperm ROS after freezing, showing the natural antioxidant property of black maca. Thus, maca appears to be a valid alternative as a cytoprotective supplement for the cryopreservation medium of epidydimal sperm of alpaca species. Further studies will be focused on the use of maca-supplemented cryostored sperm on fertility capacities using either in vitro (IVF) or in vivo (AI) models.

## Figures and Tables

**Figure 1 animals-13-02054-f001:**
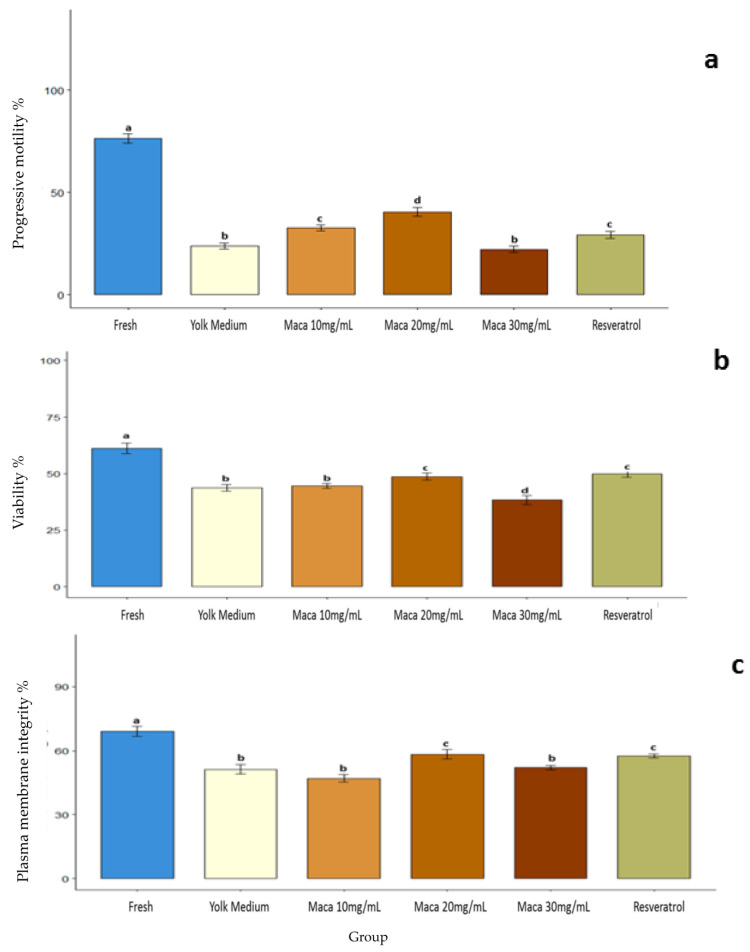
Effect of the supplementation of the freezing medium with maca (*Lepidium meyenii*) or resveratrol on sperm motility, viability, and plasma membrane integrity of epidydimal alpaca (*Vicugna pacos*) spermatozoa. (**a**) Progressive motility, (**b**) sperm viability, and (**c**) plasma membrane integrity. Fresh: raw sperm; yolk medium: sperm cryopreserved in yolk medium (YM); maca 10 mg/mL: YM supplemented with maca 10 mg/mL; maca 20 mg/mL: YM supplemented with maca 20 mg/mL; maca 30 mg/mL: YM supplemented with maca 30 mg/mL; and resveratrol: YM supplemented with resveratrol 5 mg/mL. a, b, c, d—Different letters denote significant differences.

**Figure 2 animals-13-02054-f002:**
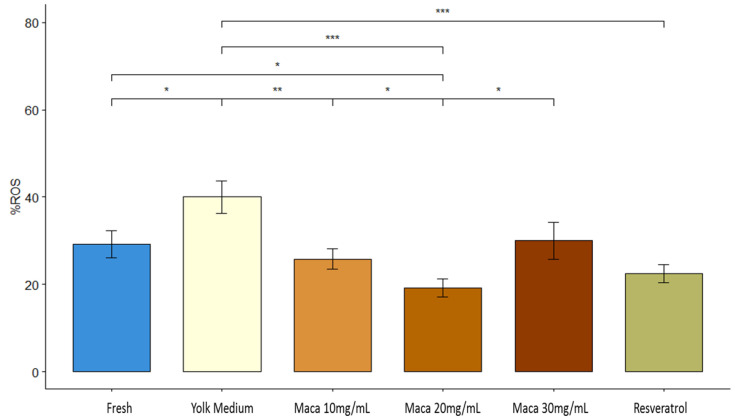
Reactive oxygen species (ROS) production assessed as the percentage of sperm positive (ROS+) for H_2_DCFDA staining. Fresh: raw sperm; yolk medium: sperm cryopreserved in yolk medium (YM); maca 10 mg/mL: YM supplemented with maca 10 mg/mL; maca 20 mg/mL: YM supplemented with maca 20 mg/mL; maca 30 mg/mL: YM supplemented with maca 30 mg/mL; and resveratrol: YM supplemented with resveratrol 5 mg/mL. Asterisks denote significant differences; *: *p* < 0.05, **: *p* < 0.01, ***: *p* < 0.001.

**Figure 3 animals-13-02054-f003:**
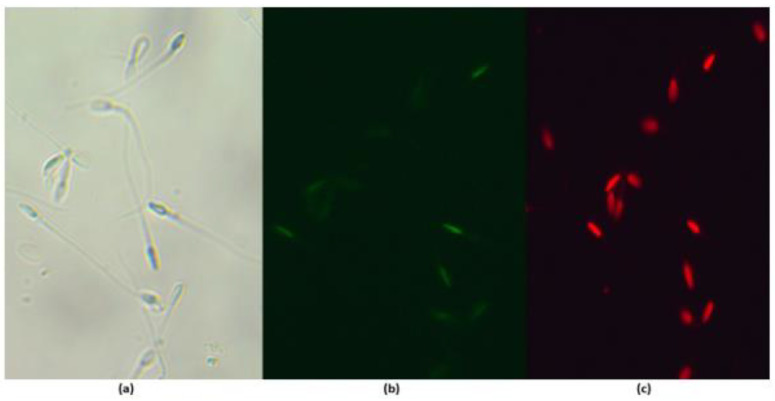
DNA fragmentation of frozen/thawed epididymal alpaca sperm. Representative images of TUNEL/propidium iodide positive (TUNEL+) spermatozoa. (**a**) Bright field; (**b**) fluorescent field depicting TUNEL-positive spermatozoa (intense green); (**c**) fluorescent field showing sperm positive for propidium iodide (red) staining. Total magnification: 400×.

**Figure 4 animals-13-02054-f004:**
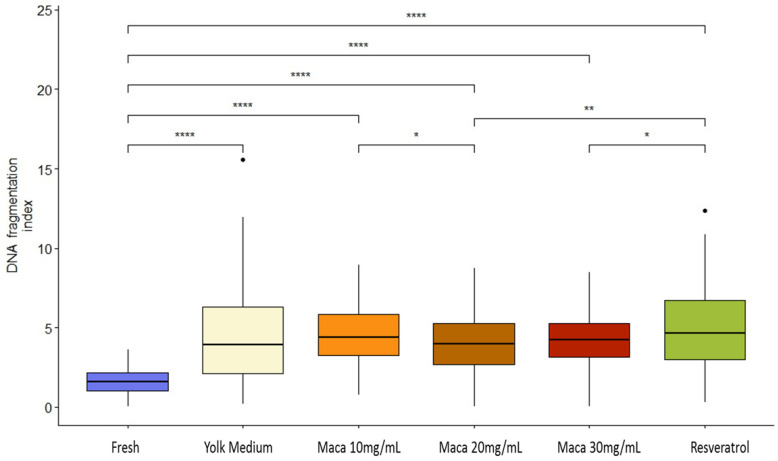
Percentage of index DNA fragmentation per treatment. Fresh: raw sperm; yolk medium: sperm cryopreserved in yolk medium (YM); maca 10 mg/mL: YM supplemented with maca 10 mg/mL; maca 20 mg/mL: YM supplemented with maca 20 mg/mL; maca 30 mg/mL: YM supplemented with maca 30 mg/mL; and resveratrol: YM supplemented with resveratrol 5 mg/mL. Asterisks denote significant differences; *: *p* < 0.05, **: *p* < 0.01, ****: *p* < 0.0001.

**Figure 5 animals-13-02054-f005:**
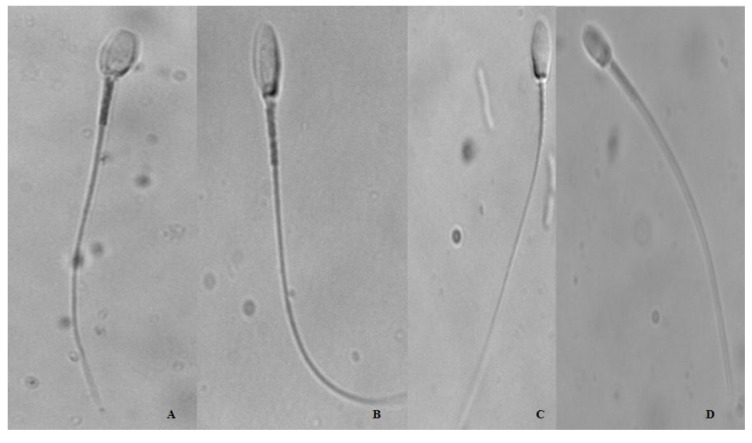
Representative images of the cytochemical reaction of endogenous reductases in the midpiece of the spermatozoon. Four types of formazan deposits were identified. (**A**) C: compact; (**B**) Sc: sub-compact; (**C**) F: focal; and (**D**) D: diffuse, in alpaca sperm mitochondria visualized after the NBT test. Photographs taken by 800× optical microscopy.

**Figure 6 animals-13-02054-f006:**
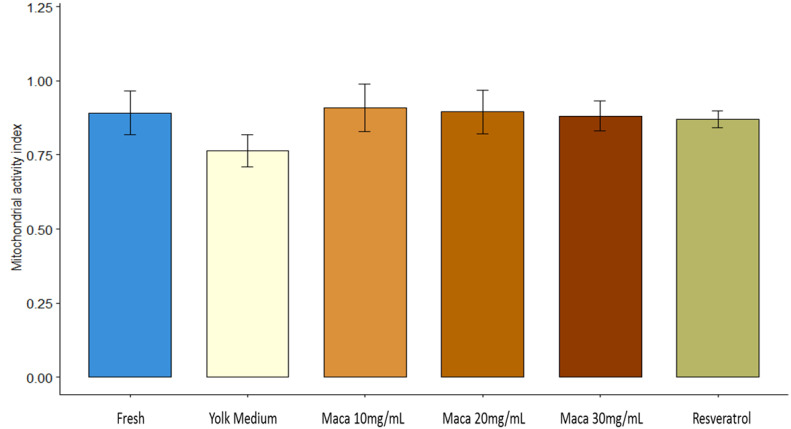
Sperm mitochondrial activity index. Data are expressed as mean ± SD of the activity index of the oxidoreductase’s enzymes in relation to the formazan deposits detected by NBT test. No significant differences were found among the groups (*p* > 0.05).

**Table 1 animals-13-02054-t001:** Rate of cytochemical activity of oxidoreductases on the fresh and frozen/thawed (*) cauda epididymis spermatozoa of alpaca species.

Rate Activity (%)						
	Compact (C)	Sub-Compact (Sub-c)	Focal (F)	Residual (Res)	Intensive Reaction (C+Sub-c) (%)	Reduced Reaction (F+Res) (%)
Fresh	38 ± 19.4	13.5 ± 9.42	0.90 ±0.65	0	51.5 ^a^	0.9 ^a^
* Yolk Medium	44.2 ± 32.8	26.0 ± 19.2	0.50 ±0.79	0.15 ± 0.28	70.2 ^b^	0.65 ^b^
* Maca 10 mg/mL	47.9 ± 32.1	13.7 ± 9.50	0.55 ± 1.50	0.01 ± 0.03	61.6 ^b^	0.55 ^c^
* Maca 20 mg/mL	21.3 ± 5.61	16.9 ± 22.4	0.38 ± 0.45	0.03 ± 0.12	38.2 ^c^	0.41 ^c^
* Maca 30 mg/mL	40.3 ± 21.2	11.0 ± 8.44	0.53 ± 0.73	0.08 ± 0.15	51.3 ^a^	0.61 ^b^
* Resveratrol	23.5 ± 12.4	11.5 ± 9.66	2.22 ± 0.46	0.02 ± 0.06	35 ^c^	2.22 ^e^

Fresh: raw sperm; yolk medium: sperm cryopreserved in yolk medium; maca 10 mg/mL: YM supplemented with maca 10 mg/mL; maca 20 mg/mL: YM supplemented with maca 20 mg/mL; maca 30 mg/mL: YM supplemented with maca 30 mg/mL; and resveratrol: YM supplemented with resveratrol 5 mg/mL. Different superscripts within the columns indicate significant differences (*p* < 0.05).

## Data Availability

Data is contained within the article or Appendix A.

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
