# Peer review of "Effect of Atomized Black Maca (Lepidium meyenii) Supplementation in the Cryopreservation of Alpaca (Vicugna pacos) Epididymal Spermatozoa"

_animals, 2023, doi:10.3390/ani13132054_

Round 1

Reviewer 1 Report

The paper testes the effect of supplementing a freezing medium with atomized black Maca powder with the objective of improving the cryopreservation outcomes of Alpaca (Vicugna pacos) epididymal sperm. 

General comments

At the introduction section the authors suggest that diluting antioxidants like ascorbic acid, methionine and alpha-tocopherol in the process of sperm cryopreservation could reduce their efficacy. To overcome this inconvenience plant extracts are presented as an alternative. However, it is not clear why are these extracts advantageous compared to other antioxidants that have been used as supplements in sperm cryopreservation protocols (lines 78-87). As results indicate, black Maca extract improves several parameters compared to the non-supplemented cryopreservation medium but when tested against resveratrol only the motility parameter shows improvement. Therefore, it seems that the resveratrol activity is not much decreased by freezing ad so it is important to better justify the alternative use of black Maca extract.

The authors test the effect of black Maca powder at three different concentrations with resveratrol as positive antioxidant control. Resveratrol is an individual molecule while the Maca extract contains  multiple substances. In lines 306-309 the authors say that "[...] the use of different cryopreservation methods, processing and types of extraction (methanol, chloroform, DMSO, and water) of Maca that can modify the concentrations of bioactive metabolites and influence its antioxidant capacity". For replication purposes the bioactive compounds of the commercial extract used should be indicated/characterised. 

Specific comments

-Line 121. "the sample was washed with PBS using centrifugation to remove tissue remains". Please indicate centrifugal force and time.

-Figures 1 and 2 (box plots) are presented only for the DNA fragmentation and ROS parameters but not for the sperm motility, viability, HOST and mitochondrial activity. 

-Please revise the formatting of references according to the journal indications.

Need some revision of text style and grammar (typographical and grammatical mistakes found).

Author Response

Dear editors,

Thank you for your letter and reviewers’ comments and constructive suggestions about our manuscript. We have read the comments carefully and made correction accordingly. In this letter, we have provided a point-to point response to each comment below.

Reviewer 1 (Anonymous)

General comments

At the introduction section the authors suggest that diluting antioxidants like ascorbic acid, methionine and alpha-tocopherol in the process of sperm cryopreservation could reduce their efficacy. To overcome this inconvenience plant extracts are presented as an alternative. However, it is not clear why are these extracts advantageous compared to other antioxidants that have been used as supplements in sperm cryopreservation protocols (lines 78-87). As results indicate, black Maca extract improves several parameters compared to the non-supplemented cryopreservation medium but when tested against resveratrol only the motility parameter shows improvement. Therefore, it seems that the resveratrol activity is not much decreased by freezing ad so it is important to better justify the alternative use of black Maca extract.

Response: Thank you for your valuable suggestion. To better justify the alternative use of black Maca, we have added the following sentence in the Introduction, Lines 106-109: “In this line, recent studies suggest the use of Maca as an alternative supplement for the preservation of semen quality in different reproductive biotechnologies, such as for sperm storage and in vitro fertilization”

The authors test the effect of black Maca powder at three different concentrations with resveratrol as positive antioxidant control. Resveratrol is an individual molecule while the Maca extract contains multiple substances. In lines 306-309 the authors say that "[...] the use of different cryopreservation methods, processing and types of extraction (methanol, chloroform, DMSO, and water) of Maca that can modify the concentrations of bioactive metabolites and influence its antioxidant capacity". For replication purposes the bioactive compounds of the commercial extract used should be indicated/characterised.

Response: Thank you for your valuable suggestion that will improve the article. For replication purposes we have included a more detailed information of the Maca extract used in this study, lines 134-140: “atomized hydroalcoholic (methanolic) extract of a commercial black Maca powder (JUVENS® Cayenatur, Lima, Peru) as described by the Research Circle of Plants with Effect on Health (Grant no. 010-2014-FONDECYT). Botanical samples were deposited in the HEPLAME MG-2015 (Herbarium of Medicinal Plants, Section of Pharmaceutical Sciences, Faculty of Sciences and Philosophy, Universidad Peruana Cayetano Heredia). Maca components have been previously characterized (Gonzales-Arimborgo 2016).”

In addition, we have modified the previous paragraph (lines 306-309) as follows in lines 407-413: “This cytoprotective effect of Maca can be associated with its antioxidant capacity. The components of black Maca have been extensively studied in recent years due to their antioxidant property. The presence of glucosinolates and macamides allows Maca (L. meyenii) to protect cells from oxidative stress damage [54,55]. Furthermore, the presence of several secondary metabolites such as macaridine, macamides, Maca alkaloids, and glucosinolates are typical in this plan [18]. These secondary metabolites are responsible for the biological and medicinal properties of the Maca.”

Specific comments

-Line 121. "the sample was washed with PBS using centrifugation to remove tissue remains". Please indicate centrifugal force 2500 RPM (Convertir a g) and time .

-Figures 1 and 2 (box plots) are presented only for the DNA fragmentation and ROS parameters but not for the sperm motility, viability, HOST and mitochondrial activity. 

-Please revise the formatting of references according to the journal indications.

Comments on the Quality of English Language

Need some revision of text style and grammar (typographical and grammatical mistakes found).

Response: We thank the reviewer for the carefully reviewing our manuscript. All Specific comments have been corrected/added to the manuscript. The manuscript has been reviewed for a native English native speaker to correct the grammar correct the grammar for a comprehensive reading.

Reviewer 2 Report

The use of natural additives to extenders to improve the quality of cryopreserved sperm is a current issue in semen conservation technology. Therefore, the presented research can be very valuable. Nevertheless, the manuscript requires supplementation and significant editorial improvement.

Below are my comments, which mainly concern the performed analyzes and presentation of results:

Please specify whether the material (testes and epididymis) was collected post mortem? This information should also be included in the methodology. How much time elapsed from the collection of the material to its transport to the laboratory? Was this time only 22 hours, or was it longer, as it may have a significant impact on the quality of sperm and their susceptibility to cryopreservation.

In the description of the methods for microscopic analysis, specify exactly what kind of microscope was used (company, manufacturer), what filter/filters were used (applies to fluorescence analyzes).

Was motility assessed in these studies using only subjective methods? (line 135-140). If so, it is difficult to assess the progressive movement in this method. Maybe it's enough to give the overall motility/percentage of motile sperm.

Please provide more details on the assessment of sperm mitochondrial activity. How the mitochondrial activity index was determined.

I have the same remark to define ROS production. How the % ROS was calculated. The description should be more detailed.

In the legends of all tables and figures, the descriptions should be improved and the significance level markings should be standardized. The authors use P˂0.05 once (e.g. line 234) and p˂=0.05 once (e.g. line 243, 254).

Did ‘*Resveratrol’ (Tables 1 and 2) contain a Mac supplement, as this can be deduced from the description under Table 1.

The table 1 has shifted rows and need to be corrected.

Does the notation ‘%ROS’ in Figure 1 mean ‘percentage of sperm with a high amount of ROS’? If so, write it down below the figure.

The notation under Figure 1 ‘Effect of Lepidium meyenii on sperm oxidative stress (ROS) activity’ is not fully understood. (Line 240).

Should ‘*YM’ be replaced with ‘*YM’ in Table 2?

In the results given as a percentage, the notation should be standardized and all values should be given to the second decimal place. (e.g. lines 285, 286).

In some places, a word is missing from the text or a sentence needs to be modified to make sense. e.g. ‘Even when other sperm parameters did decrease, no difference was detected. significant increase in lipid peroxidation.’ (lines 343-344).

Should the sentence ‘This section is not mandatory but may be added if there are patents resulting from the work reported in this manuscript.’ be in the conclusion? (line 360-361).

Please check the text carefully.

Author Response

Dear editors,

Thank you for your letter and reviewers’ comments and constructive suggestions about our manuscript. We have read the comments carefully and made correction accordingly. In this letter, we have provided a point-to point response to each comment below.

Reviewer 2 (Anonymous)

The use of natural additives to extenders to improve the quality of cryopreserved sperm is a current issue in semen conservation technology. Therefore, the presented research can be very valuable. Nevertheless, the manuscript requires supplementation and significant editorial improvement.

Below are my comments, which mainly concern the performed analyzes and presentation of results:

Please specify whether the material (testes and epididymis) was collected post mortem? This information should also be included in the methodology. How much time elapsed from the collection of the material to its transport to the laboratory? Was this time only 22 hours, or was it longer, as it may have a significant impact on the quality of sperm and their susceptibility to cryopreservation.

Response: We thank the reviewer for the carefully reviewing our manuscript and constructive suggestions that will improve the paper. The samples were obtained post-mortem and only those samples that arrived at the laboratory in 22 h or less were included in the study. Line 116-119: “testicles and epididymis from males between 4 to 6 years old were obtained postmortem from the local slaughterhouse, Huancavelica (3000-3700 m.a.s.l) Peru. The samples were stored in 0, 9% NaCl at 10°C and transported during 22 hours to the laboratory.”

In the description of the methods for microscopic analysis, specify exactly what kind of microscope was used (company, manufacturer), what filter/filters were used (applies to fluorescence analyzes).

Response: We thank the reviewer for the carefully reviewing our manuscript. This information is now detailed.

Was motility assessed in these studies using only subjective methods? (line 135-140). If so, it is difficult to assess the progressive movement in this method. Maybe it's enough to give the overall motility/percentage of motile sperm.

Response: We thank the reviewer for the carefully reviewing our manuscript. Yes, the motility was assessed using only subjective methods. We have added more information regarding this assessment, Lines 152-153: “Sperm motility was evaluated subjectively according to the published guidelines of the World Health Organization [38], as previously reported by our group (Bravo 2018)”.

Please provide more details on the assessment of sperm mitochondrial activity. How the mitochondrial activity index was determined.

Response: We thank the reviewer for the carefully reviewing our manuscript. We have added more detailed about the assessment of sperm mitochondrial activity as follows Lines 214-224: “Mitochondrial activity and the functionality index for each sample were made based on the classification of formazan deposit was determined according to the classification in percentages, obtained from the number of spermatozoas classified in each category (Compact: C,” Sub-compact: “SC,” Focal: “F,” Residual: “R”) and their corresponding factor (1, 0.7, 0.3, 0.1, respectively) proposed by Hrudka [42]. Cytochemical activity rates were determined by using a semi-quantitative rating according to the amount of formazan deposited in the mitochondrial sheath, where the Compact (strong and uniform staining) and Subcompact (medium to strong deposit), both were classified as Intensive Reactions. On the other hand, Focal (few localized deposits) and Diffuse (weak and/or absent reaction) patterns were related to Low and Residual rates as reduced reactions”.

I have the same remark to define ROS production. How the % ROS was calculated. The description should be more detailed.

Response: We thank the reviewer for the carefully reviewing our manuscript. We have added more detailed about the assessment of percentage of ROS+ sperm as it follows: Lines 202-207: “After staining, the samples were kept in a dark chamber and evaluated by fluorescence microscopy (EUROMEX, Zeiss Axio Scope A,Holland). Fluorescence was detected by using a 530/30 nm bandwidth filter. The results are expressed as the percentage of cells positive to Hâ‚‚DCFDA (ROS+) staining based on the count of 100 spermatozoa for each sample.”

In the legends of all tables and figures, the descriptions should be improved and the significance level markings should be standardized. The authors use P˂0.05 once (e.g. line 234) and p˂=0.05 once (e.g. line 243, 254).

Response: Thank you for your comments that have improved the quality of our manuscript. We have corrected this information across all the manuscript.

Did ‘*Resveratrol’ (Tables 1 and 2) contain a Mac supplement, as this can be deduced from the description under Table 1.

Response: We thank the reviewer for the carefully reviewing our manuscript. No, the “Resveratrol group” did not contain Mac supplement. To clarify it, we have modified the methodology as it follows, Lines141-142: “In addition, the supplementation of the YM freezing medium with 5 mg/ml of Resveratrol (Sigma, St. Louis, MO, USA) was included as an antioxidant control group”.

The table 1 has shifted rows and need to be corrected.

Does the notation ‘%ROS’ in Figure 1 mean ‘percentage of sperm with a high amount of ROS’? If so, write it down below the figure.

The notation under Figure 1 ‘Effect of Lepidium meyenii on sperm oxidative stress (ROS) activity’ is not fully understood. (Line 240).

Should ‘*YM’ be replaced with ‘*YM’ in Table 2? Si es Yolk medium mantener la nomenclatura

In the results given as a percentage, the notation should be standardized and all values should be given to the second decimal place. (e.g. lines 285, 286).

In some places, a word is missing from the text or a sentence needs to be modified to make sense. e.g. ‘Even when other sperm parameters did decrease, no difference was detected. significant increase in lipid peroxidation.’ (lines 343-344).

Should the sentence ‘This section is not mandatory but may be added if there are patents resulting from the work reported in this manuscript.’ be in the conclusion? (line 360-361).

Please check the text carefully.

Response: We thank the reviewer for the carefully reviewing our manuscript. All these specific comments have been corrected/added to the manuscript. The manuscript has been reviewed for a native English native speaker to correct the grammar correct the grammar for a comprehensive reading.

Reviewer 3 Report

The authors evaluate the effect of the addition of atomized black Maca in the cryopreservation medium of epididymal spermatozoa of alpaca species. The spermatozoa from male alpaca adults are known to have bad freezability potential and all the efforts to improve alpaca semen preservation are of interest for the artificial insemination field.

The authors showed that alpaca sperm quality was significantly affected by the cryopreservation process and that the supplementing the cryopreservation medium with black Maca at 20 mg/ml improved sperm parameters and decreases the total emission of ROS. They correctly demonstrate the interest of black Maca due to antioxidant capacity in alpaca sperm cryopreservation.

The design of the study is correct, but manuscript including methods and discussion need rewriting (see below). The discussion may be completed for a comprehensive reading of the potential of their results.

Clearly, the work fit the journal scope and the question is original. The results provide an advancement of the current knowledge about alpaca semen cryopreservation. The manuscript should be then accepted after major revision (see main comments and point out text modifications requirement).

Major concerns:

The authors wrote that they are investigated marker associated functionality and cellular markers (line 32). The authors would have improved their manuscript by discussing sperm protein markers, already used in artificial insemination market, including those related to motility and freezability potential.

Actually, such marker may could have been also investigated if the authors still have the samples especially since they will be novel in applications for alpaca sperm preservation. See for instance:

Duracka M, Benko F and Tvrdá E (2023) Molecular Markers: A New Paradigm in the Prediction of Sperm Freezability. Int. J. Mol. Sci. Vol. 24:3379. https://doi.org/10.3390/ijms24043379

Among the sperm protein markers, proAKAP4 marker is a sperm protein that is related to sperm motility, male fertility and freezability that have been described in main mammals such as bulls, stallions, and camelids (review in Carracedo et al. 2022 ; Malo et al. 2021), and recently use to investigate at the molecular level the supplementation of extenders for cryopreservation in bulls and stallions (Blommaert et al. 2021 ; Kowalsky et al. 2022).

Kowalczyk A, Gałeska E and Bubel A (2022) The concentration of proAKAP4 and other indicators of cryopotential of spermatozoa cryopreserved in extender with holothuroidea extract addition. Animals. Vol. 12:521. https:// doi.org/10.3390/ani12040521

Carracedo S, Briand-Amirat L, Dordas-Perpinyà M, Ramos Escuredo Y, Delcombel R, Sergeant N and Delehedde M (2022) ProAKAP4 protein marker: towards a functional approach to male fertility. Animal Reproduction Science. Vol.247(107074):1-20.https://doi.org/10.1016/j.anireprosci.2022.107074

Blommaert D, Sergeant N, Delehedde M, Donnay I, Lejeune JP, Franck T and Serteyn D (2021) First results about ProAKAP4 concentration in stallion semen after cryopreservation in two different freezing media. Cryobiology. Vol. 102:133-135

Below, subsection analysis of the submitted manuscript.

Abstracts

Both abstracts should be rewrite as there is need for editing and corrections.

For instance, there is missing verb in the sentence in Line 31 – 32: “Markers associated with functionality such as motility, vitality and plasma membrane integrity.”

Introduction

Introduction is clear, relevant for the field and presented in a well-structured manner.

Materials and Methods

This section is well organized and with appropriate references. The manuscript’s results will be mainly reproducible based on the details given. Main comments on this section are below.

Line 109: Please confirm alpaca origin: Vicugna pacos (such as the title ?)

Section 2.2 - Line 117: can the authors comment about the step : 15 minutes at 37°C “to recover”. Is there a reference?

Section 2.2 - Line 119: the authors should add more details about the “two parts” of the sample (how they performed) and mentioned clearly the two parts. Please rewrite.

The pronoun “It” was to many times used in sub-section 2.4.2.  and 2.4.5. Please rewrite.

Line 129 : please add “Lepidium meyenii”as referenced in the Results.

Results

Results are clear and well presented. The figures and tables are appropriate and properly show the data. Main comments on this section are below.

The “3.1. Effect of Maca on sperm parameters” is only an introducing sentence of the following results.

Please begin numbering after this introducing sentence.

Line 228: Add “Table 1 and Figure 1” since the values commented are those of the Table 1.

Discussion

There is a sentence with no sense line 323 “significant increase in lipid peroxidation [49].” Please complete.

Line 344 and 345: please check this paragraph.

As said before presentation of sperm protein marker will be interesting in this manuscript. Discussion will be improved if the authors will comment on sperm functional protein marker, especially those that have been recently introduced for large camelids (Malo et al. 2021) and that are sperm freezability marker relative to oxidative stress (Duracka et al 2023 ; Nixon et al. 2019 ; Riesco et al. 2020 ; Carracedo et al. 2022).

Riesco M, Anel-Lopez L, Neila-Montero M, Palacin-Martinez C, Montes-Garrido R, Alvarez M, de Paz P, Anel L (2020) ProAKAP4 as Novel Molecular Marker of Sperm Quality in Ram: An Integrative Study in Fresh, Cooled and Cryopreserved Sperm. Biomolecules. 10(7):1046. doi: 10.3390/biom10071046.

Nixon B, Bernstein I, Cafe SL, Delehedde M, Sergeant S, Eamens AL, Lord T, Dun MD, De Iuliis GN and Bromfield EG (2019b) A Kinase Anchor Protein 4 is vulnerable to oxidative adduction in male germ cells. Frontiers in Cell and Developmental Biology – Vol. 7:319.

Conclusions

The conclusions are consistent with the evidence.

References

The authors should follow the instructions to authors. Especially in the list of the authors (respect the adding of comma and/or points). Also check out the Journal (italic or not italic).

Minor comments

The authors will need a complete read of the manuscript to improve spelling and facilitate the reading.

Line 32: verb is missing.

Line 48:  “s”

Line 54: need a comma instead of a point.

Line 72 and line 77: repetitive word (however)

Line 206: spelling: integrity

Through all the manuscript: check out the way the authors are writing concentrations as this should be“ mg/mL” (with a space after the number)

Example: write 20 mg/mL instead of “20mg/ml”

Line 221: (Table 1)

Line 228-232: Justify the paragraph.

Line 232: Resveratrol

Line 236: concentrations

Line 243: Resveratrol

Line 291: Comma to be removed: a,b,c,d,e

Table 2 check spelling of Standard.

Line 342: 5°C

Line 323: Check out : Even when other sperm parameters did decrease, no difference was 343 detected. significant increase in lipid peroxidation [49].

Line 350: italic for in vitro and in vivo

Line 359: subsection Patent

Please check out spelling and writing

Author Response

Dear editors,

Thank you for your letter and reviewers’ comments and constructive suggestions about our manuscript. We have read the comments carefully and made correction accordingly. In this letter, we have provided a point-to point response to each comment below.

Reviewer 3 (Anonymous)

The authors evaluate the effect of the addition of atomized black Maca in the cryopreservation medium of epididymal spermatozoa of alpaca species. The spermatozoa from male alpaca adults are known to have bad freezability potential and all the efforts to improve alpaca semen preservation are of interest for the artificial insemination field.

The authors showed that alpaca sperm quality was significantly affected by the cryopreservation process and that the supplementing the cryopreservation medium with black Maca at 20 mg/ml improved sperm parameters and decreases the total emission of ROS. They correctly demonstrate the interest of black Maca due to antioxidant capacity in alpaca sperm cryopreservation.

The design of the study is correct, but manuscript including methods and discussion need rewriting (see below). The discussion may be completed for a comprehensive reading of the potential of their results.

Clearly, the work fit the journal scope and the question is original. The results provide an advancement of the current knowledge about alpaca semen cryopreservation. The manuscript should be then accepted after major revision (see main comments and point out text modifications requirement).

 Response: We thank the reviewer for the valuable comments that improve the quality of our manuscript. The methods and discussion sections has been rewritten and reviewed for a native English native speaker to correct the grammar for a comprehensive reading.

Major concerns:

The authors wrote that they are investigated marker associated functionality and cellular markers (line 32). The authors would have improved their manuscript by discussing sperm protein markers, already used in artificial insemination market, including those related to motility and freezability potential.

Response: Thank you for your valuable review that will improve the article. We have included your suggestion in the discussion section.

Actually, such marker may could have been also investigated if the authors still have the samples especially since they will be novel in applications for alpaca sperm preservation. See for instance: Duracka M, Benko F and Tvrdá E (2023) Molecular Markers: A New Paradigm in the Prediction of Sperm Freezability. Int. J. Mol. Sci. Vol. 24:3379. https://doi.org/10.3390/ijms24043379.

Among the sperm protein markers, proAKAP4 marker is a sperm protein that is related to sperm motility, male fertility and freezability that have been described in main mammals such as bulls, stallions, and camelids (review in Carracedo et al. 2022 ; Malo et al. 2021), and recently use to investigate at the molecular level the supplementation of extenders for cryopreservation in bulls and stallions (Blommaert et al. 2021 ; Kowalsky et al. 2022).

Kowalczyk A, Gałeska E and Bubel A (2022) The concentration of proAKAP4 and other indicators of cryopotential of spermatozoa cryopreserved in extender with holothuroidea extract addition. Animals. Vol. 12:521. https:// doi.org/10.3390/ani12040521

Carracedo S, Briand-Amirat L, Dordas-Perpinyà M, Ramos Escuredo Y, Delcombel R, Sergeant N and Delehedde M (2022) ProAKAP4 protein marker: towards a functional approach to male fertility. Animal Reproduction Science. Vol.247(107074):1-20.https://doi.org/10.1016/j.anireprosci.2022.107074

Blommaert D, Sergeant N, Delehedde M, Donnay I, Lejeune JP, Franck T and Serteyn D (2021) First results about ProAKAP4 concentration in stallion semen after cryopreservation in two different freezing media. Cryobiology. Vol. 102:133-135

Response: Thank you for your valuable suggestion. Unfortunately, we cannot include this analysis in the present article, however we will consider this approach for future investigations.

Below, subsection analysis of the submitted manuscript.

Abstracts

Both abstracts should be rewrite as there is need for editing and corrections. For instance, there is missing verb in the sentence in Line 31 – 32: “Markers associated with functionality such as motility, vitality and plasma membrane integrity.”

Response: We thank the reviewer for the carefully reviewing our manuscript. We have corrected both abstracts according to your suggestions.

Introduction

Introduction is clear, relevant for the field and presented in a well-structured manner.

Response: We thank the valuable comment.

Materials and Methods

This section is well organized and with appropriate references. The manuscript’s results will be mainly reproducible based on the details given. Main comments on this section are below.

Line 109: Please confirm alpaca origin: Vicugna pacos (such as the title?)

Response: We thank the reviewer for the carefully cheking our manuscript. This was corrected.

Section 2.2 - Line 117: can the authors comment about the step: 15 minutes at 37°C “to recover”. Is there a reference?

Response: We thank the reviewer for the carefully cheeking our manuscript. We have modified this sentence to make it clearer as it follows, Line 123-125: “Then, once the samples were extracted, the suspension containing the spermatozoa was transferred to 1.5 ml plastic tubes and placed in an incubator to warm up to 37°C. Subsequently, the samples were divided into two equal parts”.

Section 2.2 - Line 119: the authors should add more details about the “two parts” of the sample (how they performed) and mentioned clearly the two parts. Please rewrite

Response: We thank the reviewer for the carefully cheeking our manuscript. We have modified this sentence to make it clearer as it follows, Line 125-128:” One part washed with PBS using centrifugation at 400xg for 5 minutes and then used to determine the motility, viability and integrity of the plasma membrane in the raw sample. The second part was used for cryopreservation”.

The pronoun “It” was to many times used in sub-section 2.4.2.  and 2.4.5. Please rewrite.

Line 129 : please add “Lepidium meyenii”as referenced in the Results.

Corrected in all manuscript 

Response: We thank the reviewer for the carefully cheeking our manuscript. We have corrected this according to your suggestions.

Results

Results are clear and well presented. The figures and tables are appropriate and properly show the data.

Response: We appreciate the valuable comment.

Main comments on this section are below.

The “3.1. Effect of Maca on sperm parameters” is only an introducing sentence of the following results.

Please begin numbering after this introducing sentence.

Line 228: Add “Table 1 and Figure 1” since the values commented are those of the Table 1.

Response: We appreciate the valuable comment. We have corrected this according to your suggestions.

Discussion

There is a sentence with no sense line 323 “significant increase in lipid peroxidation [49].” Please complete.

Line 344 and 345: please check this paragraph.

Response: We thank the reviewer for the carefully reviewing our manuscript. All these specific comments have been corrected/added to the manuscript. The manuscript has been reviewed for a native English native speaker to correct the grammar and make it easier to read and understand.

As said before presentation of sperm protein marker will be interesting in this manuscript. Discussion will be improved if the authors will comment on sperm functional protein marker, especially those that have been recently introduced for large camelids (Malo et al. 2021) and that are sperm freezability marker relative to oxidative stress (Duracka et al 2023 ; Nixon et al. 2019 ; Riesco et al. 2020 ; Carracedo et al. 2022).

Response: Thank you for your valuable review that will improve the article. We have included your suggestion in the discussion section.

Conclusions

The conclusions are consistent with the evidence.

Response: We appreciate the valuable comment.

References

The authors should follow the instructions to authors. Especially in the list of the authors (respect the adding of comma and/or points). Also check out the Journal (italic or not italic).

Response: We thank the reviewer for the carefully cheeking our manuscript. This was corrected according to your suggestion.

Minor comments

The authors will need a complete read of the manuscript to improve spelling and facilitate the reading.

Line 32: verb is missing.

Line 48:  “s”

Line 54: need a comma instead of a point.

Line 72 and line 77: repetitive word (however)

Line 206: spelling: integrity

Through all the manuscript: check out the way the authors are writing concentrations as this should be“ mg/mL” (with a space after the number)

Example: write 20 mg/mL instead of “20mg/ml”

Line 221: (Table 1)

Line 228-232: Justify the paragraph.

Line 232: Resveratrol

Line 236: concentrations

Line 243: Resveratrol

Line 291: Comma to be removed: a,b,c,d,e

Table 2 check spelling of Standard.

Line 342: 5°C

Line 323: Check out : Even when other sperm parameters did decrease, no difference was 343 detected. significant increase in lipid peroxidation [49].

Line 350: italic for in vitro and in vivo

Line 359: subsection Patent

Comments on the Quality of English Language

Please check out spelling and writing

Response: We thank the reviewer for the carefully reviewing our manuscript. All these specific/minor comments have been corrected. The manuscript has been reviewed for a native English native speaker to correct the grammar and make it easier to read and understand.

Round 2

Reviewer 2 Report

Dear Authors,

The manuscript has been significantly improved and I recommend it for publication.

Reviewer 3 Report

Thanks to the author to article improvement. 

I recommend to accept in the present form.